# *Prunus spinosa* Extract Sensitized HCT116 Spheroids to 5-Fluorouracil Toxicity, Inhibiting Autophagy

**DOI:** 10.3390/ijms232416098

**Published:** 2022-12-17

**Authors:** Maria Condello, Rosa Vona, Stefania Meschini

**Affiliations:** 1National Center for Drug Research and Evaluation, National Institute of Health, 00161 Rome, Italy; 2Center for Gender-Specific Medicine, National Institute of Health, 00161 Rome, Italy

**Keywords:** colorectal cancer, spheroids, drug resistance, *Prunus spinosa*, 5-fluorouracil, apoptosis, autophagy

## Abstract

Autophagy is a lysosomal degradation and recycling process involved in tumor progression and drug resistance. The aim of this work was to inhibit autophagy and increase apoptosis in a 3D model of human colorectal cancer by combined treatment with our patented natural product *Prunus spinosa* + nutraceutical activator complex (PsT + NAC^®^) and 5-fluorouracil (5-FU). By means of cytotoxic evaluation (MTT assay), cytofluorimetric analysis, light and fluorescence microscopy investigation and Western blotting evaluation of the molecular pathway PI3/AKT/mTOR, Caspase-9, Caspase-3, Beclin1, p62 and LC3, we demonstrated that the combination PsT + NAC^®^ and 5-FU significantly reduces autophagy by increasing the apoptotic phenomenon. These results demonstrate the importance of using non-toxic natural compounds to improve the therapeutic efficacy and reduce the side effects induced by conventional drugs in human colon cancer.

## 1. Introduction

Colorectal cancer (CRC) was the second leading cause of cancer death in Europe in 2020 (245,000 deaths) and the second most commonly diagnosed cancer (520,000 new cases) [1]. Despite the remarkable success of new antitumor drug treatments, 5-fluorouracil (5-FU) has remained the main component of combination protocols for the treatment of CRC since the 1990s.

The chemotherapy drug 5-FU is a synthetic analogue of fluorinated pyrimidine, which has been extensively studied and used not only in the treatment of CRC but also for cancer of the respiratory, digestive and breast systems [2]. Moreover, 5 -FU is converted in cancer cells into active metabolites, such as fluorodeoxyuridine monophosphate (FdUMP). FdUMP is able to inhibit the action of the enzyme thymidylate synthase (TYMS), thus reducing the rate of DNA synthesis. 

Specifically, 5-FU in combination with leucovorin is the standard treatment for early-stage CRC, and it is applied in combination with oxaliplatin (FOLFOX regimens) or irinotecan (FOLFIRI) for patients with metastatic CRC [3]. Moreover, 5-FU has also been used in combination with target drugs, such as vascular endothelial growth factor (VEGF) inhibitors (bevacizumab, ramucirumab) and epidermal growth factor receptor (EGFR) anti-receptors (cetuximab and panitumumab) [4]. Unfortunately, complications and resistance mechanisms occurring after 5-FU chemotherapy severely limit its use.

Chemotherapy, after an initial positive effect, produces disease relapses in many patients, mainly due to the triggering of resistance to the drugs used (multidrug resistance, MDR). Both innate and acquired chemoresistance represent obstacles to cancer eradication [5]. The molecular mechanisms contributing to this phenomenon are numerous and interrelated [6,7]. Many studies in recent years have demonstrated an important role of autophagy in the development of the drug resistance mechanism [8,9]. In this regard, many authors show that inhibition of the autophagy mechanism can lead to an increase in toxicity, with a consequent reduction in resistance, in tumors treated with different chemotherapeutic compounds [10,11]. Autophagy is a phenomenon related to the lysosomal degradation of proteins, ribosomes and cell organelles, is triggered by metabolic stresses and is involved in complex signaling networks that determine cell fate [12]. After drug treatment, cancer cells activate autophagy by degrading damaged organelles or recycling amino acids or fatty acids through the formation of autophagosomes. This adaptation mechanism, which promotes cell survival, tumor growth and chemoresistance, is involved in CRC resistance to 5-FU [13]. Several studies have shown that the inhibition of autophagy enhances the antitumor effect of 5-FU on CRC [14]. In vitro and in vivo studies on CRC have shown that autophagy inhibitors such as as chloroquine enhance the cytotoxic effect of 5-FU and inhibit tumor growth [15,16]. Zhang and colleagues demonstrated that treating human colon carcinoma cells with curcumin and 5-FU significantly attenuated autophagy and promoted the cytotoxicity of the drug [17]. 

The aim of our work was to assess whether our patented *Prunus spinosa* Trigno ecotype (PsT) extract was able to sensitize CRC to the drug 5-FU by modulating autophagic resistance. Previous studies have shown that *Prunus* sp. flower extract has a pro-oxidant and toxic effect on hepatocarcinoma cells [18]. The methanolic extract of *Prunus* sp. fruits was evaluated for its toxic effect on glioblastoma multiforme and pancreatic cancer cell lines. The results showed that the extract had antioxidant capabilities and reduced the viability of glioblastoma cells [19]. Our previous study showed that *Prunus* sp. Trigno ecotype drupe extract with a nutraceutical activator complex (NAC), consisting of amino acids, vitamins and mineral salt mixtures, inhibited the growth of human colon carcinoma cell lines (HCT116 and SW480), cervical carcinoma cell lines (HeLa) and human bronchoalveolar adenocarcinoma cells (A549) [20]. No effect was shown on normal cells, small intestinal epithelial cells (IEC-6) and gingival fibroblasts [21]. Our studies were then continued in 3D colon cancer in vitro and in vivo models. We compared the effect of a high concentration (P10 mg/mL) of *Prunus* sp. complex plus NAC (Trigno M, commercial compound) with 5-FU alone on 3D cells, demonstrating the reduced efficacy of 5-FU alone and the efficacy of the *Prunus* sp. complex plus NAC. Furthermore, Trigno M was shown to delay the growth of CRC in the animal model [22].

There are no data in the literature reporting the effect of combining *Prunus* sp. extract with conventional chemotherapeutics. Therefore, in this work, we analyzed the complex PsT + NAC^®^ in combination with 5-FU on a 3D model of colon carcinoma cells. We evaluated the chemosensitizing effect of the combined treatment by MTT assay, morphological changes by light microscopy, actin fiber changes by fluorescence, as well as apoptosis induction by flow cytometry and Western blotting. In addition, investigation of the molecular pathway (AKT/mTOR) and proteins related to the establishment of the autophagy phenomenon (BECN1, p62, LC3 I/II) showed that the *Prunus* sp. plus NAC complex is able to sensitize CRC to 5-FU by significantly reducing chemotherapy-induced survival autophagy.

## 2. Results

### 2.1. Chemosensitizing Effect of PsT + NAC^®^ with 5-FU

We first verified the chemosensitizing effect of PsT + NAC^®^ on the 3D CRC model. Figure 1 shows the cell viability percentage obtained with the MTT assay after single and combined treatments with PsT + NAC^®^ and 5-FU. 

Cell viability data for treatment with NAC alone were not reported, because we verified the possible toxicity of NAC in a previous study [21]. Cell viability did not change after treatment with 5-FU alone (approximately 92%). Cell viability decreased in a dose-dependent manner after treatment with different concentrations of *Prunus* sp. alone (approximately 88.5% after PsT 2 mg/mL + NAC^®^, approximately 77.2% after PsT 4 mg/mL + NAC^®^, 73.2% after PsT 6 mg/mL + NAC^®^). When PsT + NAC^®^ was administered in combination with 5-FU, there was a greater reduction in cell viability (81.8%, 72.3% and 62.6%, respectively) than with PsT + NAC^®^ or 5-FU alone.

These results show that PsT + NAC^®^ is able to sensitize HCT116 cells to 5-FU chemotherapy, overcoming drug resistance.

### 2.2. Morphological Observations by Optical and Scanning Electron Microscopy

Untreated and treated HCT116 spheroids were observed under a light microscope (Figure 2).

Untreated HCT116 spheroids showed a compact structure with a well-defined border; their shape resembled the organization of intestinal loops (Figure 2A). Treatment with 5-FU alone, or PsT 2 mg/mL + NAC^®^ (alone or in combination), did not change the morphology of the spheroids, which were similar to the control (Figure 2B–D). After treatment with PsT 4 mg/mL + NAC^®^, alone or in combination with 5-FU, the spheroids began to disintegrate; the membrane, which was intact and defined, was no longer organized and single rounded cells were observed (Figure 2E,F). Notably, after the combined treatment, the large spheroid disintegrated into smaller, less dense and compact spheroids.

This effect was much more evident when spheroids were treated with PsT 6 mg/mL + NAC^®^, alone or in combination with the chemotherapy drug (Figure 3A,B).

The morphological changes were confirmed by scanning electron microscopic observations: the compact shape of the control spheroid (Figure 3C) was lost when the spheroid was treated with PsT 6 mg/mL + NAC^®^ alone (Figure 3D). Furthermore, single cells began to detach from the spheroid mass when the combined treatment PsT 6 mg/mL + NAC^®^ and 5-FU was performed (Figure 3E).

### 2.3. Remodeling of Cell Morphology and Cytoskeletal Actin

The morphological changes observed after combined treatment were mainly due to actin alterations caused by cellular remodeling (Figure 4).

As previously demonstrated in Figure 2A for the light microscopy, the observations made under the fluorescence microscope (Figure 4A–C) also showed that the untreated spheroids had regular and compact edges, indicating a well-organized actin that maintained cellular regularity. After treatment with 5-FU, no particular differences were observed compared to the control sample (Figure 4D–F). After treatment with only PsT 6 mg/mL + NAC^®^, the nuclei maintained their morphology, but the actin began to thicken and condense (Figure 4G–I). The thickening and the change in regular actin assembly were even more evident after treatment with PsT 6 mg/mL + NAC^®^ and 5-FU, where the release of actin fragments was also observed in the detaching cells (Figure 4L).

Overall, these observations demonstrated that the morphological changes induced by the combined treatment were due to the rearrangement of the actin cytoskeleton.

### 2.4. Combined Treatment Induces Apoptosis in 3D Colon Cancer Cells

The apoptotic induction of the combination treatment on the 3D colon cancer model was demonstrated by Annexin V-FITC/PI flow cytometric analysis (Figure 5). The data are reported in Figure 5 and Table 1.

Treatment with 5-FU (300 µM for 24 h) alone slightly increased the percentage of apoptotic cells, especially the late apoptotic fraction (30.5 ± 1.5% compared to 13.0 ± 1.0% in the untreated sample). According to MTT data, when increasing the concentration of PsT + NAC^®^ alone, the percentage of apoptotic cells increased (21.5% after PsT 2 mg/mL + NAC^®^, 27.5% after PsT 4 mg/mL + NAC^®^ and 51.0% after PsT 6 mg/mL + NAC^®^ for 24 h, compared to 20.5% of control cells). After combined PsT + NAC^®^ and 5-FU treatment, the apoptotic rate increased (31.0% after PsT 2 mg/mL + NAC^®^ and 5-FU, 48.5% after PsT 4 mg/mL + NAC^®^ and 5-FU and 60.5% after PsT 6 mg/mL + NAC^®^ and 5-FU) when compared to 5-FU (39.5%) or PsT + NAC^®^ alone. These results showed that PsT + NAC^®^ sensitized HCT116 spheroids to the 5-FU drug by inducing apoptosis.

The apoptotic effects of combined treatment (PsT + NAC^®^ and 5-FU) were also demonstrated by Western blotting analysis of the caspases’ expression (Figure 6).

In particular, increasing doses of PsT + NAC^®^ and 5-FU treatment led to a significant increase in the expression of Caspase-9 (Figure 6A) and Capsase-3 (Figure 6B). The data, obtained by Western botting analysis, show that this increase was statistically significant compared to single treatment with PsT + NAC^®^ and 5-FU or control (*p* ≤ 0.01). It should be noted also that the maximum dose of PsT + NAC^®^ and 5-FU treatment induced a significant reduction in Caspase-9 levels (*p* ≤ 0.01). Our findings indicate that the combined treatment, PsT + NAC^®^ and 5-FU, clearly induces apoptosis in human colorectal carcinoma cell line HCT116.

### 2.5. PsT + NAC^®^ Treatment Affects the Autophagic Pathway

We also investigated whether treatment with PsT + NAC^®^ could influence the autophagy process via one of the most important regulatory pathways of autophagy, the PI3K/AKT/mTOR signaling pathway.

Using Western blot analysis, we assessed the levels of AKT and phospho-AKT (Figure 7A,B). We found that the amount of AKT (Figure 7A) increased when spheroids were treated with 5-FU or increasing doses of PsT + NAC^®^ alone. In contrast, the highest doses of combined treatment (6 mg/mL PsT + NAC^®^ and 5-FU) induced a significant reduction in AKT levels compared with PsT + NAC^®^ alone or untreated spheroids (*p* ≤ 0.05). Furthermore, by analyzing the phosphorylation and activity of AKT (Figure 7B), we found that treatment with PsT + NAC^®^ significantly increased, in a dose-dependent manner compared to untreated spheroids, the phosphorylation of the protein, and, interestingly, this increase persisted in the combined treatment (PsT + NAC^®^ and 5-FU). It is known in the literature that the phosphorylation of AKT is able to promote the phosphorylation of mTOR, the main suppressor of autophagy. Therefore, we wished to investigate whether treatment with PsT + NAC^®^ could affect both molecular pathways. The amount of mTOR protein increased significantly when spheroids were treated with PsT 6 mg/mL + NAC^®^ alone (*p* ≤ 0.05) or in combination with 5-FU (*p* ≤ 0.01), compared with untreated spheroids (Figure 7C). Similarly, the expression of its phosphorylated form (phospho-mTOR) increased significantly (*p* ≤ 0.01) after treatment with the highest dose of PsT + NAC^®^, while, in combined treatment, there was a significant increase (*p* ≤ 0.01) in a dose-dependent manner (Figure 7D).

To confirm the effects of PsT + NAC^®^ treatment on autophagy, we also examined the expression status of some autophagy-related proteins involved in the different steps of this process, such as Beclin1 (BECN1), p62 and LC3s. As shown in Figure 7E, the levels of BECN1 were significantly increased (*p* ≤ 0.05) after treatment with PsT + NAC^®^, compared with untreated spheroids; this increase occurred in a dose-independent manner and was also present in the combined treatment. The amount of p62 (Figure 7F) increased significantly (*p* ≤ 0.05) after treatment with the highest dose of PsT + NAC^®^ (PsT 6 mg/mL + NAC^®^), compared with untreated spheroids, and maintained a consistently high value even in the combined treatments. The highest dose of PsT + NAC^®^ resulted in a significant increase (*p* ≤ 0.05) in p62 levels compared with both untreated and single-treated spheroids (Figure 7F). The expression value of LC3 II increased after PsT + NAC^®^ treatment (PsT 2 mg/mL + NAC^®^, PsT 4 mg/mL + NAC^®^) compared with untreated spheroids, but decreased significantly (*p* ≤ 0.05) at the highest concentration (PsT 6 mg/mL + NAC^®^) (Figure 7G). It is noteworthy that the same trend of LC3 II was also observed in the combined treatment (PsT6 + NAC^®^ + 5-FU), with a significant decrease compared to the spheroids treated with the lowest doses of PsT + NAC^®^. These results demonstrate how the autophagy process is inhibited by PsT + NAC^®^ when used in combination with the chemotherapeutic 5-FU. In spheroids obtained from human colorectal carcinoma cells, treatment with the natural product may play a key role in reducing the dose of 5-FU chemotherapy and improving side effects due to drug therapy.

## 3. Discussion

The drug 5-FU is an antimetabolite drug widely used in the treatment of different types of cancer, such as breast and colorectal cancer (CRC). Unfortunately, the first diagnosis for most patients is made late, and survival with a single regimen in an advanced disease stage is only 10–15% [13], increasing to 40–50% in combined-regimen treatment [23]. Therefore, it is essential to conduct further studies to investigate the mechanisms underlying the failure of 5-FU chemotherapy. The clinical application of 5-FU is limited by the development of drug resistance mechanisms after chemotherapy. The emergence of tumor cells with a multidrug resistance (MDR) phenotype leads to increased cell tolerance to chemotherapeutics and survival strategies.

Apoptosis and autophagy are two important cellular regulatory events that can lead to tumor cell death or, in the case of autophagy, survival phenomena and thus promote tumor growth, triggering the MDR phenotype. When tumor cells are under active replicative and metabolic stress, they can adopt the autophagy mechanism and sequester damaged organelles and proteins in lysosomes, thereby inducing a protective catabolic mechanism to avoid therapy-induced damage and maintain cellular homeostasis [24]. With 5-FU treatment, the autophagic survival response is activated as a resistance mechanism. Over the years, many pharmacological attempts have been made to increase the sensitivity of colon cancer cells, but, often, the high toxicity and relative lack of specificity have directed research towards combination studies with biologically active, natural food compounds [25,26]. Our previous work has shown that the natural product PsT plus NAC^®^, whose peculiar composition is specified in Italian Patent No. RM2015A 000133, 4 January 2015, has an antiproliferative and antitumor effect on the HCT116 colon adenocarcinoma line [21,22,27]. This effect was observed on 3D lines obtained by us on the same cell line and inoculated in vivo on a mouse model. These studies showed that the cytotoxic and cytostatic effect of the compound alone is comparable to that obtained with 5-FU alone. Therefore, we analyzed the combination of the two compounds PsT plus NAC^®^ and 5-FU, using them at lower concentrations (PsT 2 mg/mL + NAC^®^, PsT 4 mg/mL + NAC^®^ mg/mL, or PsT 6 mg/mL + NAC^®^ plus 5-FU 300 µM for 24 h), to facilitate the evaluation of the chemosensitizing effect of the natural product.

As shown in the cell viability assay performed on the 3D line, the combination of PsT plus NAC^®^ and 5-FU reduced cell viability in all combinations compared with 5-FU alone (Figure 1). A viability-reducing effect was also demonstrated with the single PsT + NAC^®^ treatments, but in a smaller percentage than with the combinations. This first experiment demonstrated the greater cytotoxic effect induced by the natural product compared to treatment with 5-FU alone. Light microscope images show the cytotoxic effect induced by treatment with PsT + NAC^®^ and with PsT + NAC^®^ and 5-FU (Figure 3A,B). The disintegration of the spheroids in the combination (Figure 3A) and the compactness and homogeneity present in the treatment with 5-FU alone (Figure 2B) are clearly visible. This effect was confirmed by SEM observations, where spheroids were disintegrated in single cells.

Microscopic observations led us to investigate the role of actin microfilaments in the combination treatments. In addition to its many functions, actin also has that of providing structural support for the plasma membrane; its fibers run longitudinally according to cell polarity and are particularly abundant at the periphery so as to define its contours [28,29]. Fluorescence microscopy observations confirmed the damage induced by the combined PsT plus NAC^®^ and 5-FU treatment at the cortical actin level under the plasma membrane, compared with single treatments (Figure 4). To verify the presence of the apoptosis phenomenon in our treatments, we performed a quantitative analysis using the Annexin V assay in cytofluorimetry. In the combined treatments, the apoptosis rate was significantly elevated, further demonstrating the enhanced cytotoxic effect induced by PsT + NAC^®^ when administered in combination with 5-FU on the 3D colon carcinoma line (Figure 5). In support of the assessment of apoptosis by cytofluorimetry, we analyzed the proteolytic cascade of Caspase-9 (initiator) and Caspase-3 (executor) by Western blotting (Figure 6). In the combined PsT + NAC^®^ and 5-FU treatment, a marked increase in the expression of Caspase-9 and Caspase-3 was observed compared with single treatments. Importantly, in the case of PsT 6 mg/mL + NAC^®^ and 5-FU, a drastic reduction in Caspase-9 and a marked and significant increase in Caspase-3, known to play a central role in the execution of apoptosis, was observed [30]. These results confirm the presence of the phenomenon of programmed cell death in the combined PsT + NAC^®^ and 5-FU treatments in the 3D colon carcinoma line.

Many cytotoxic drugs, including 5-FU, activate apoptosis but also autophagy as a resistance mechanism [23]. Some authors reported that 5-FU-induced DNA damage led to LC3-II and other autophagy proteins’ upregulation, inducing nucleophagy (a form of autophagy) in microsatellite-stable CRC cell lines, leading to resistance to 5-FU treatment [31,32]. In addition, the antitumor enhancing effect of 5-FU by chloroquine has been amply demonstrated both in vitro and in vivo models. Authors demonstrated in animals that chloroquine inhibited the autophagic process by restoring the sensitivity condition [16]. With autophagy, cancer cells have the opportunity to eliminate unfolded proteins and damaged organelles, thus benefiting from the generation of glycolytic substrates [33]. Among the various factors that can lead to the onset of cancer, one that plays an important role is the alteration of the PI3K/AKT/mTOR pathway [34]. In the evaluation of AKT and its phosphorylated form, an increase in the amount of AKT was observed with the single treatments (5-FU or PsT + NAC^®^, Figure 7A), as well as a significant reduction in the combined treatment (PsT + NAC^®^ and 5-FU). In contrast, the phosphorylated activity of AKT (Figure 7B) increased significantly with PsT + NAC^®^ treatment compared to untreated spheroids, and the increased expression persisted even in the combined treatment (PsT + NAC^®^ and 5-FU). Once activated, AKT phosphorylates many substrates, including mTOR, one of the best-known downstream effectors of AKT. It has recently been shown that mTOR is able to mediate anti-cancer drug resistance by suppressing autophagy [35]. In our study, we showed that the phosphorylated form of mTOR increased significantly when spheroids were treated with PsT 6 mg/mL + NAC^®^ alone or in combination with 5-FU, compared with the control (Figure 7C). This result demonstrates the important role of mTOR in the inhibition of protective autophagy.

Continuing our analysis, we examined the expression of some proteins involved in different steps of the autophagy mechanism, Beclin1 (BECN1), p62 and LC3s (Figure 7E–G). BECN1 is a critical regulator of autophagy in CRC metastasis. In CRC patients, downregulation of BECN1 significantly promoted the motility and invasion of CRC cells [36]. The authors showed the significantly lower expression of BECN1 in CRC tumors compared with adjacent normal colon tissue, and BECN1 downregulation was correlated in patients with poor prognosis. Guo et al. showed, in 68 patients with colorectal cancer, a direct correlation between the overexpression of BECN1 and LC3 in tumor tissue compared to normal tissue [37]. Koukourakis. et al., on 155 patients, confirmed that the overexpression of BECN1 plays a key role in the aggressiveness of colon cancer. Of course, there are also papers demonstrating the dual role of BECN1, correlating both its over- and under-regulation with the poorer survival of patients undergoing drug therapies [38].

In our 3D model, BECN1 expression increased after all treatments compared with control cells and increased with PsT + NAC^®^ and 5-FU (Figure 7E). It is known that autophagy and apoptosis are related cellular mechanisms. When autophagy is deficient or inhibited, apoptosis is induced, and vice versa—autophagy is induced when apoptosis is deficient [39]. Therefore, the overexpression of BECN1 can suppress proliferation, invasion and induce apoptosis in our model of colon cancer. SQSTM1 (sequestosome 1, p62) is an autophagy protein that binds to ubiquitinated protein aggregates and transports them into autophagosomes [40]. The expression of p62 (Figure 7F) is significantly increased after PsT + NAC^®^ treatment compared to control cells, with a further increase after combined treatment (PsT + NAC^®^ and 5-FU). Keisuke Kosumi et al. demonstrated that p62 protein degradation occurs in colon carcinoma when the autophagy mechanism is active [41]. In our model, increased p62 levels indicate a blockage of the autophagic process by PsT + NAC^®^, an effect that increases when combined with 5-FU. Another important player in the autophagic process is the cytoplasmic form of microtubule-associated protein 3 (LC3-I, 16 kDa), which is activated, transferred and converted to the phosphatidylethanolamine (PE)-conjugated form, LC3-II (14 kDa), which is membrane-associated and finally recruited to the autophagosomes [12]. The expression of LC3 II (Figure 7G) increased after PsT + NAC^®^ treatments, to decrease significantly at the highest concentration (PsT 6 mg/mL + NAC^®^) and especially after the combined treatment (PsT 6 mg/mL + NAC^®^ and 5-FU). These results demonstrate a clear decrease in autophagy when colon cancer spheroids are treated with the combination of the natural product, PsT 6 mg/mL + NAC^®^, with the chemotherapeutic agent, 5-FU.

Altogether, our findings indicate that treatment with PsT + NAC^®^ and 5-FU induces a profound change in the balance between autophagy and apoptotic in the 3D colon carcinoma line.

## 4. Materials and Methods

### 4.1. Plant Material

Blackthorn fruits of *Prunus spinosa* Trigno ecotype (PsT) were collected in the Molise region, in South-Central Italy, characterized by a typical Mediterranean climate. The fruits were collected and stored in refrigerated bags. The PsT extraction was performed by macerating the vegetable material in a water/alcohol solvent (60° of alcohol) for several days. The drying process was performed by conventional methods for evaporation under reduced pressure, spray drying or lyophilization. We obtained a solution with a concentration of 86 mg/mL (PsT 86 mg/mL). The phenolic acid, flavone/ol and anthocyanin content was analyzed according the protocols reported by Meschini et al., 2017 [21].

### 4.2. Three-Dimensional Cell Cultures

The established human colorectal carcinoma cell line (HCT116) was provided by the American Type Culture Collection (ATCC, Manassas, VA, USA). The spheroid model was obtained by seeding HCT116 cells (1.4 × 10^5^ cells) on ultra-low-attachment 6-multiwell plates (Corning Costar, Cambridge, MA, USA) and culturing them in RPMI medium (Gibco Life Technologies, Paisley, UK), supplemented with L-glutamine, 10% FBS and 1% penicillin (50 U/mL)–streptomycin (50 µg/mL), in a humidified atmosphere at 37 °C, 5% CO_2_ for 72 h. The spheroids obtained had a variable size between 100 and 300 µm.

### 4.3. Treatments

Cells were treated with different solutions obtained after the progressive dilution of PsT 86 mg/mL solution with a complex blend of amino acids, vitamins and minerals, called the nutraceutical activator complex (NAC) [20]. We used PsT 2 mg/mL + NAC^®^, PsT 4 mg/mL + NAC^®^ and PsT 6 mg/mL + NAC^®^ for 24 h.

Then, 5-fluorouracil (5-FU, Sigma Aldrich, Saint Louis, MO, USA) 300 µM, a fluoro-pyrimidine analogue widely employed for the treatment of colon cancer, was used in combination with different concentrations of PsT + NAC^®^.

### 4.4. MTT Assay

Cell viability was assessed by the 3-(4,5-dimethylthiazol-2-yl)-2,5-diphenyltetrazolium bromide (MTT) assay (Sigma Aldrich), with slight modifications to the standard protocol [42].

After removing the cell medium with mild centrifugation, untreated and treated samples were washed with phosphate-buffered saline (PBS, Sigma) and incubated with 1 mg/mL MTT solution for 2 h at 37 °C. After removing the MTT solution with centrifugation, the samples were lysed by dimethyl sulfoxide (DMSO) and analyzed with a microplate reader (Bio-Rad, Hercules, CA, USA) at 570 nm. Cell viability (%) was calculated as follows: (absorbance mean value of the treated sample/absorbance mean value of the control sample) ×100.

### 4.5. Optical Microscopy

Spheroids were photographed with an inverted microscope (ECLIPSE Ti2 light microscope) equipped with a CCD camera (Nikon Europe, Amsterdam, The Netherlands).

### 4.6. Scanning Electron Microscopy (SEM)

In order to evaluate the morphological characteristics and alterations induced by single or combined treatments, SEM analysis was conducted on HCT116 spheroids. After 24 h of treatment, the spheroids were deposited on circular polylysinated slides for 24 h and then fixed with 2.5% glutaraldehyde plus 2% sucrose in 0.1 M cacodylate buffer (pH 7.4) at room temperature for 1 h. After washing twice with the same buffer, the samples were post-fixed in 1% osmium tetroxide for 90 min at room temperature and dehydrated through a graded ethanol solution. Then, they were dried with hexamethyldisilazane for 30 min and gold-coated by sputtering (SCD040 Balzers device, Baltec, Leichtenstein, Germany). The samples were examined with a FEI Quanta Inspect FEG field emission scanning electron microscope (FEI Company, Eindhoven, The Netherlands).

### 4.7. Immunostaining and Fluorescence Microscopy

After treatment, all steps were performed in Eppendorf tubes. Samples were fixed with 4% paraformaldehyde plus 2% sucrose solution for 15 min at room temperature. After washing with PBS, samples were permeabilized with 0.3% Triton X-100 for 15 min. This was followed by incubation with solution block in PBS:0.1% BSA, 0.2% Triton X-100, 0.05% Tween-20, 10% goat serum, for 1 h at room temperature. The samples were incubated with anti-actin antibody conjugated with fluorescein isothiocyanate (FITC) in block solution overnight at 4 °C. Finally, after washing with PBS, nuclei were stained with Hoechst33342 solution (1:1000 dilution, Sigma Aldrich) for 30 min at 37 °C. After washing with PBS, pellets were mounted on a microscope slide with PBS:glycerol (1:1). Samples were observed using an ECLIPSE Ti2 light microscope (Nikon Europe, Amsterdam, Netherlands).

### 4.8. Quantification of Apoptosis on Spheroids by Annexin V-FITC Labelling

An Annexin V-FITC/propidium iodide (PI) apoptosis detection kit (eBioscence, San Diego, CA, USA) was used to investigate cell death and apoptotic induction. All samples were disaggregated into single cells with a syringe and washed with binding buffer solution (1 X). Cells were incubated with Annexin V-FITC (5 µL in 100 µL of cell suspension) for 15 min and then with PI solution (5 µL in 200 µL of cell suspension). Samples were analyzed with a BDLSRII flow cytometer (Becton, Dickinson & Company, Franklin Lakes, NJ, USA) equipped with a 5 mW, 488 nm, air-cooled argon ion laser and a Kimmon HeCd 325 nm laser. The fluorescence emission was obtained through a 530 nm band pass filter for FITC and a 575 nm band pass filter for PI. At least 10,000 events/sample were acquired in log mode for Annexin V-FITC/PI labeling. Percentages of apoptotic, necrotic and viable cells were calculated using the FACS Diva Software (Becton, Dickinson & Company).

### 4.9. Western Blotting

The whole-cell extract was obtained in RIPA buffer (50 mM Tris-HCl, pH = 7.4, 150 mM NaCl, 1% NP-40, 2 mM EDTA plus 10 µg/mL, 0.1% SDS, 50 mM NAF) in the presence of standard protease and phosphatase inhibitors at + 4 °C, according to standard procedures. The protein content was determined with a protein assay reagent (Bio-Rad Laboratories Inc., PA, USA), using bovine serum albumin as a standard. Equal protein content of total cell lysates (30µg) was resolved on 10% or 12% SDS-PAGE and electrically transferred onto poly(vinylidene difluoride) membranes (Bio-Rad Laboratories, Hercules, CA, USA). Membranes were blocked with TBS-T (20 mM Tris-HCl pH 7.4, 150 mM NaCl, 0.02% Tween-20) containing 5% skimmed milk (Bio-Rad Laboratories), for 1 h at room temperature, and then incubated overnight at 4 °C with primary antibodies diluted in TBS-T containing 5% milk or 5% BSA. MAbs: anti-Caspase-9 (Cell Signaling Technology, Inc., Beverly, MA, USA; # 9508S; dil. 1:1000); anti-BECN1 (Santa Cruz; # sc-48381; dil. 1:1000); anti-mTOR (Cell Signaling Technology, Inc., Beverly, MA, USA; #4517S; dil. 1:1000); pAb: anti-p-Akt (Cell Signaling Technology; #9271S; dil. 1:1000); anti-LC3 (Novus Biological, #NB100-2220; 1:1000); anti-Akt (Santa Cruz; # sc-8312; dil. 1:1000); anti-Caspase-3 (Cell Signaling; # 9661S; dil. 1:1000); anti-SQSTM1/p62 (Sigma-Aldrich; #P0067; dil. 1:1000); anti-p-mTOR (Cell Signaling Technology, Inc.; # 2971S; dil. 1:1000); primary antibodies were used. After three washes in TBS-T, immune complexes were detected with horseradish-peroxidase-conjugated species-specific secondary antibodies (Jackson Laboratory, Bar Harbor, ME, USA). Membranes were developed using ECL detection reagents (Millipore Corporation, Billerica, MA, USA). Reactive bands were detected by the ChemiDocMP system (Bio-Rad Laboratories Inc.) and signal quantification was performed using the IMAGE LAB software (Bio-Rad). The arbitrary units obtained were used to calculate the relative increase/decrease in bands. To ensure the presence of equal amounts of protein, the membranes were re-probed with anti-GAPDH (Santa Cruz; #sc-32233; dil. 1:2000).

### 4.10. Statistical Analyses

The results obtained from three independent experiments were expressed as mean ± standard deviation. One-way analysis of variance (ANOVA) and Bonferroni post hoc analysis were applied to reveal differences between all samples, using the GraphPad Prism 5 software (GraphPad, San Diego, CA, USA). The alpha level was set at *p* < 0.05; each figure legend specifies symbols and significance.

## 5. Conclusions

The autophagic mechanism in CRC has a complex role depending on intratumor genetic heterogeneity. Autophagy promotes tumor progression under conditions of cellular stress. In the literature, many authors agree with our research, stating that the increased expression of autophagy proteins in CRC is linked to metastatic progression and poor prognosis.

In our work, we show that autophagy is an essential process for cancer cells to overcome the metabolic perturbations induced by drug therapy. Furthermore, the suppression of autophagy, induced by treatment with the natural product PsT + NAC^®^ and 5-FU, on the 3D human colon carcinoma cell line (HCT116) is necessary and sufficient to generate a metabolic vulnerability that leads 3D cells to an energy crisis and apoptosis. This work also aimed to emphasize the importance of further investigations into therapeutic combinations between conventional drugs and natural products that could lead to a better understanding of resistance mechanisms and reduced drug dosing, as well as helping to reduce adverse health effects.

## Figures and Tables

**Figure 1 ijms-23-16098-f001:**
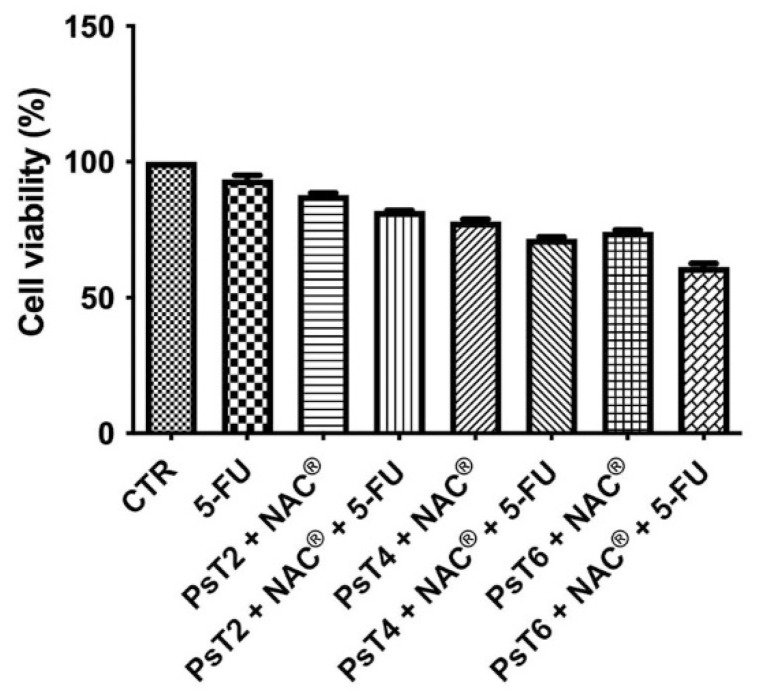
Cell viability of HCT116 spheroids. The MTT test was performed after treatment with 5-FU alone (300 µM), PsT 2 mg/mL + NAC^®^, PsT 4 mg/mL + NAC^®^, PsT 6 mg/mL + NAC^®^ alone or in combination with 5-FU for 24 h. The results, obtained from three independent experiments, each in sixfold, are expressed as mean ± standard deviation; *: *p* < 0.05 vs. control; #: *p* < 0.05 vs. each single treatment.

**Figure 2 ijms-23-16098-f002:**
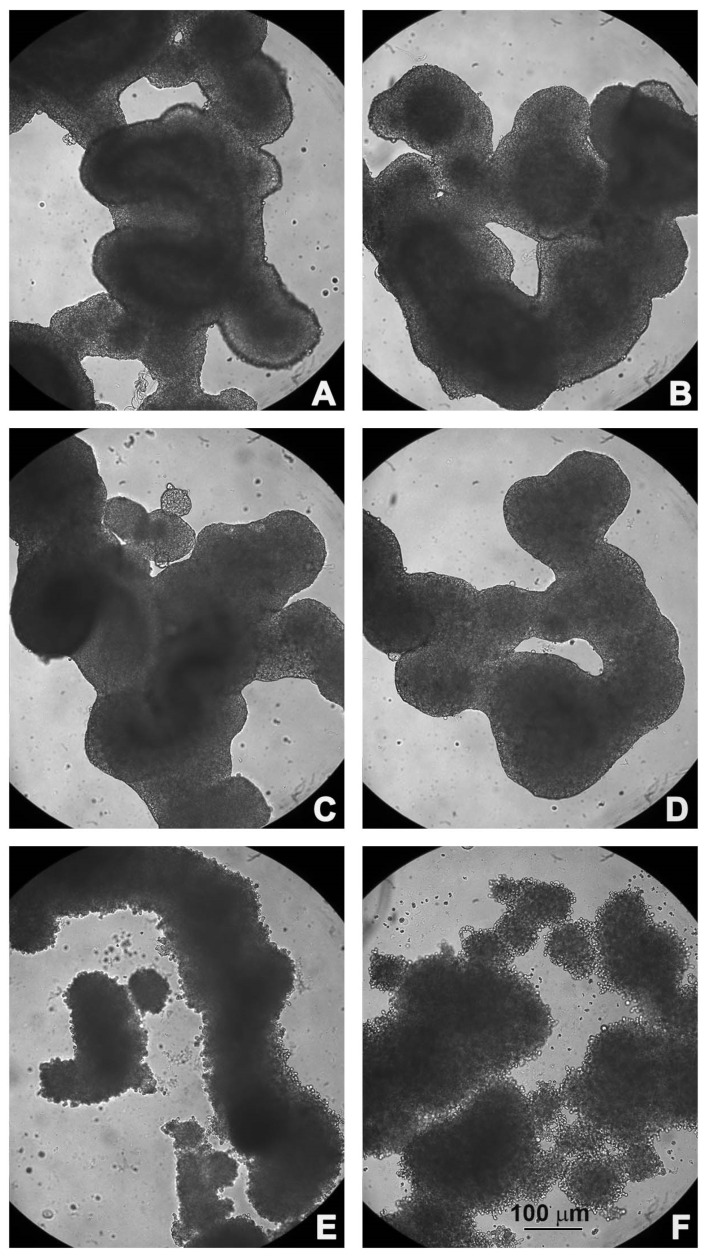
Morphological observations under light microscopy. (**A**) Untreated HCT116 spheroids; (**B**) treated with 5-FU; (**C**) PsT 2 mg/mL + NAC^®^; (**D**) PsT 2 mg/mL + NAC^®^ and 5-FU; (**E**) PsT 4 mg/mL + NAC^®^; (**F**) PsT 4 mg/mL + NAC^®^ and 5-FU for 24 h.

**Figure 3 ijms-23-16098-f003:**
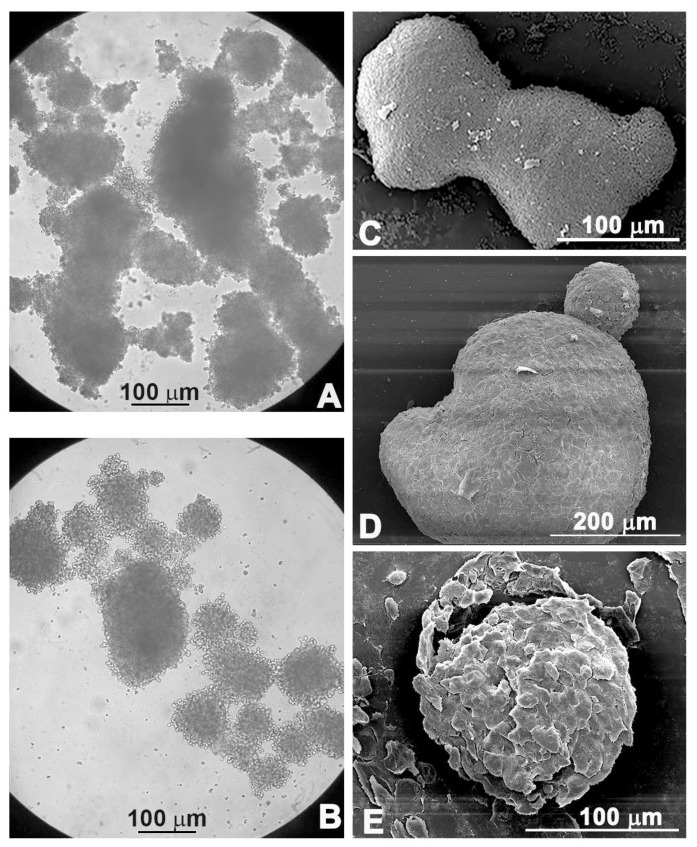
Morphological observations under light microscopy (**A**,**B**) and scanning electron microscopy (**C**–**E**). Untreated spheroids (**C**), spheroids treated with PsT 6 mg/mL + NAC^®^; (**A**,**D**) treated with PsT 6 mg/mL + NAC^®^ and 5-FU for 24 h (**B**,**E**).

**Figure 4 ijms-23-16098-f004:**
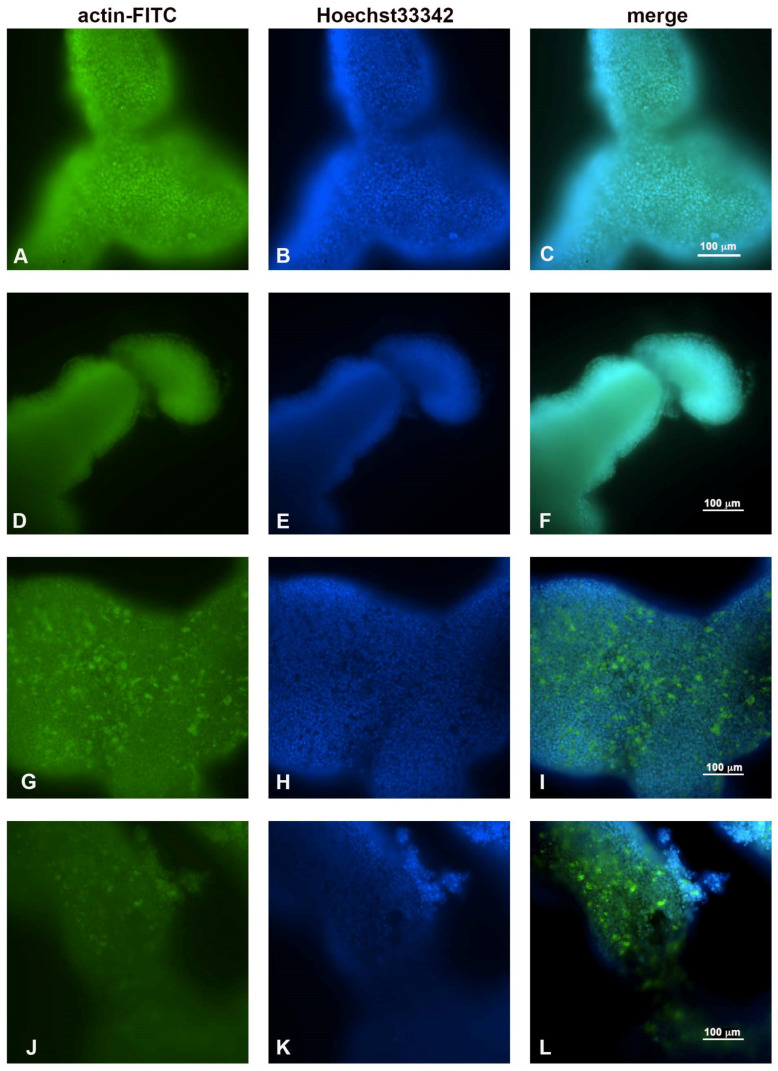
Fluorescence microscopic observations of actin alterations. (**A**,**B**,**C**) Untreated HCT116 spheroids; (**D**,**E**,**F**) treated with 5-FU for 24 h; (**G**,**H**,**I**) PsT 6 mg/mL + NAC^®^; (**J**,**K**,**L**) PsT 6 mg/mL + NAC^®^ and 5-FU for 24 h. Scale bar: 100 µm.

**Figure 5 ijms-23-16098-f005:**
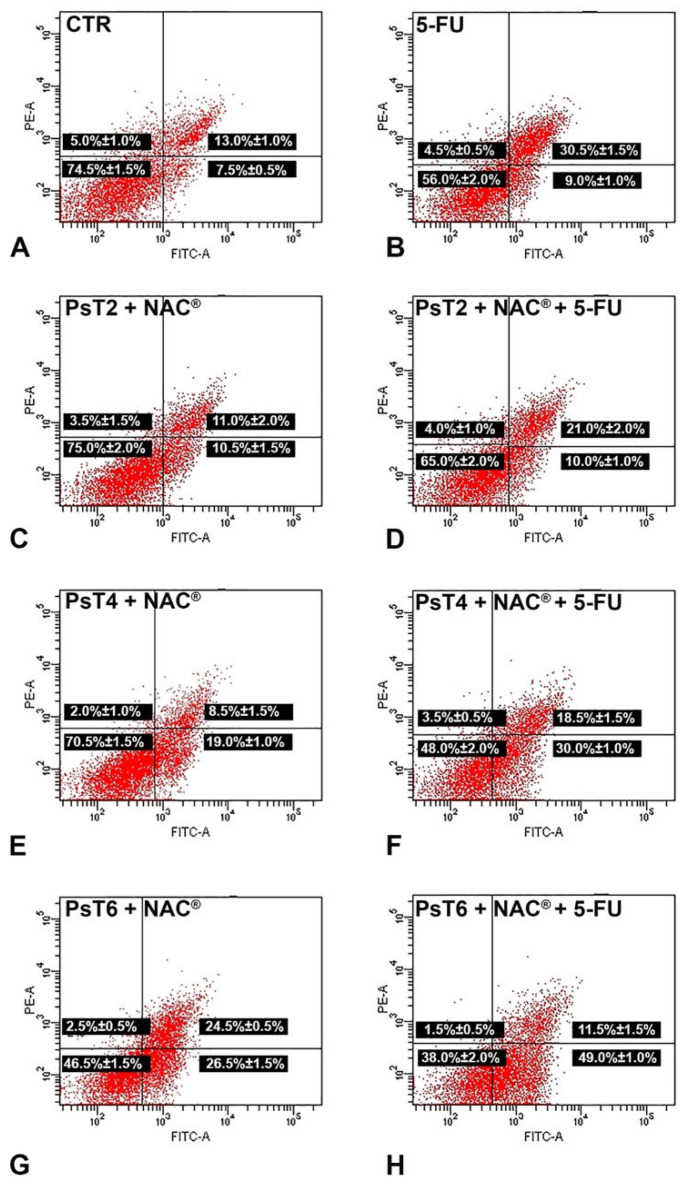
Representative dot plots of apoptosis induction on HCT116 spheroids after Annexin V/PI dual staining. (**A**) Control spheroids; (**B**) treated with 5-FU; (**C**) PsT 2 mg/mL + NAC^®^; (**D**) PsT 2 mg/mL + NAC^®^ and 5-FU; (**E**) PsT 4 mg/mL + NAC^®^; (**F**) PsT 4 mg/mL + NAC^®^ and 5-FU; (**G**) PsT 6 mg/mL + NAC^®^; (**H**) PsT 6 mg/mL + NAC^®^ and 5-FU for 24 h. Dot plots are representative of three independent experiments; values indicate mean ± standard deviation. For each dot plot: the lower left quadrant shows viable cells; the lower right quadrant shows early apoptotic cells; the upper right quadrant shows late apoptotic cells; the upper left quadrant shows necrotic cells. FITC-A: fluorescein isothiocyanate signal. PE-A: propidium iodide signal.

**Figure 6 ijms-23-16098-f006:**
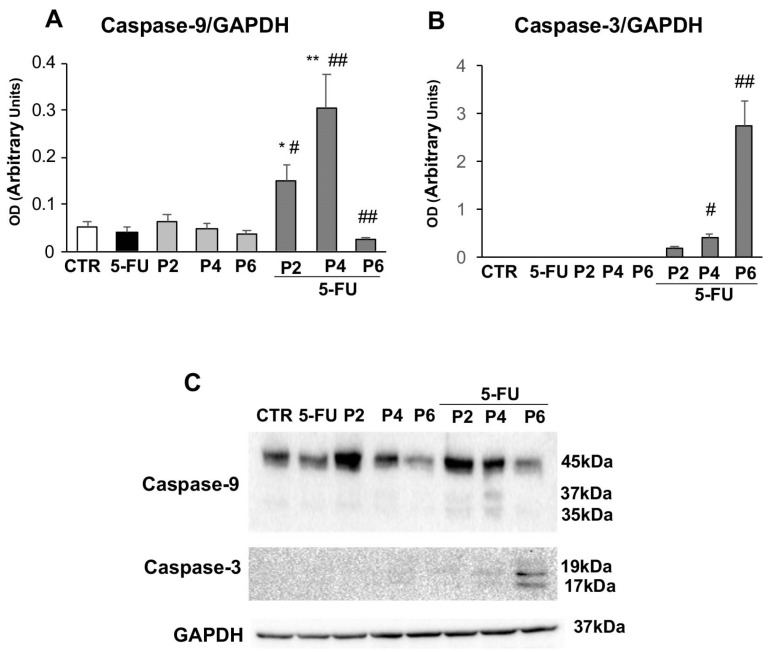
Apoptosis analysis. The bar graph shows the relative densitometric quantification of Caspase-9 (**A**) and Caspase-3 (**B**) in 3D human colorectal carcinoma cell line HCT116. Each protein is normalized to GAPDH. Data are expressed as mean ± SD of three independent experiments. * *p* ≤ 0.05 vs. control; ** *p* ≤ 0.01 vs. control; # *p* ≤ 0.05 vs. PsT + NAC^®^ or 5-FU treatment alone; ## *p* ≤ 0.01 vs. PsT + NAC^®^ or 5-FU treatments alone. (**C**) Representative Western blot analysis of Caspase-9 and -3. The determination of GAPDH was used as a loading control.

**Figure 7 ijms-23-16098-f007:**
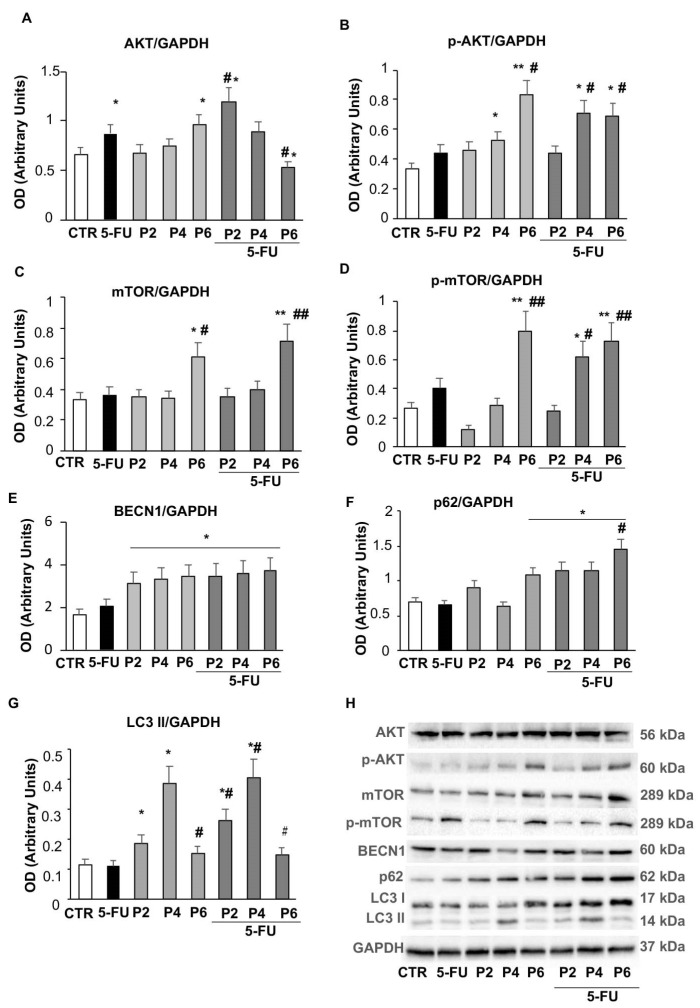
Autophagy pathway. Bar graphs show the relative densitometric quantification of AKT (**A**), phospho-AKT (**B**), mTOR (**C**), phospho-mTOR (**D**), BECN1 (**E**), p62 (**F**) and LC3 (**G**) in 3D colorectal carcinoma cell lines. Each protein is normalized to GAPDH. Data are expressed as mean ± SD of three independent experiments. * *p* ≤ 0.05 vs. control; ** *p* ≤ 0.01 vs. control; # *p* ≤ 0.05 vs. 5-FU or PsT + NAC^®^ treatment alone; ## *p* ≤ 0.01 vs. 5-FU or PsT + NAC^®^ treatment alone. (**H**) Representative Western blot analysis of all proteins. Determination of GAPDH was used as a loading control.

**Table 1 ijms-23-16098-t001:** Results of Annexin V/PI dual staining of HCT116 spheroids after single and combined treatment.

	Viable Cells	Early Apoptotic Cells	Late Apoptotic Cells	Necrotic Cells
CTR	74.5 ± 1.5%	7.5 ± 0.5%	13.0 ± 1.0%	5.0 ± 1.0%
5-FU	56.0 ± 2.0%	9.0 ± 1.0%	30.5 ± 1.5%	4.5 ± 0.5%
PsT 2 mg/mL + NAC^®^	75.0 ± 2.0%	10.5 ± 1.5%	11.0 ± 2.0%	3.5 ± 1.5%
PsT 2 mg/mL + NAC^®^ + 5-FU	65.0 ± 2.0%	10.0 ± 1.0%	21.0 ± 2.0%	4.0 ± 1.0%
PsT 4 mg/mL + NAC^®^	70.5 ± 1.5%	19.0 ± 1.0%	8.5 ± 1.5%	2.0 ± 1.0%
PsT 4 mg/mL + NAC^®^ + 5-FU	48.0 ± 2.0%	30.0 ± 1.0%	18.5 ± 1.5%	3.5 ± 0.5%
PsT 6 mg/mL + NAC^®^	46.5 ± 1.5%	26.5 ± 1.5%	24.5 ± 0.5%	2.5 ± 0.5%
PsT 6 mg/mL + NAC^®^ + 5-FU	38.0 ± 2.0%	49.0 ± 1.0%	11.5 ± 1.5%	1.5 ± 0.5%

## Data Availability

Not applicable.

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
