# Peer review of "Prunus spinosa Extract Sensitized HCT116 Spheroids to 5-Fluorouracil Toxicity, Inhibiting Autophagy"

_ijms, 2022, doi:10.3390/ijms232416098_

Round 1

Reviewer 1 Report

The authors of the article presented a comprehensive analysis of the impact of Prunus Spinosa+Nutraceutical Activator Complex with 5-Fluorouacil (5-FU) chemotherapeutic drug for colorectal cancer cells. The presented results allow to state that the proposed form of anti -cancer therapy has significant potential and can substantially contribute to improve treatment efficiency.

The presented article after minor changes should be approved for publication.

Minor revisions

1. The form of the presented data in Figure 4 is not legible. I understand that people using Annexinv-Fitc/Pi Flow Cyttometric  Analysis will not have a problem with understanding the graphic presentation, however, the presented data should be understandable to a wider audience.

In my opinion, to improve the readability of this data, the meaning of the abbreviations describing individual axes should be clearly exploited. Unambiguous criteria for fluorescent signals indicating apoptotic cells should be defined.

Another aspect is the size of the charts used, which only in the electronic version after enlargement allows you to read numerical quantities.

2.  The size of Figure 5 should also be changed so that the presented data is clear, scaling used in Figure 6 would be more adequate.

Author Response

The authors of the article presented a comprehensive analysis of the impact of Prunus Spinosa+Nutraceutical Activator Complex with 5-Fluorouacil (5-FU) chemotherapeutic drug for colorectal cancer cells. The presented results allow to state that the proposed form of anti -cancer therapy has significant potential and can substantially contribute to improve treatment efficiency.

The presented article after minor changes should be approved for publication.

Minor revisions

  1. The form of the presented data in Figure 4 is not legible. I understand that people using Annexinv-Fitc/Pi Flow Cyttometric  Analysis will not have a problem with understanding the graphic presentation, however, the presented data should be understandable to a wider audience. In my opinion, to improve the readability of this data, the meaning of the abbreviations describing individual axes should be clearly exploited. Unambiguous criteria for fluorescent signals indicating apoptotic cells should be defined. Another aspect is the size of the charts used, which only in the electronic version after enlargement allows you to read numerical quantities.

Figure 4 has become figure 5. To improve the readability of the flow cytometric data, we have clarified the meaning of the abbreviations (FITC-A and PE-A) and dot plot quadrants in the figure legend. We enlarged the font of the percentages shown in the dot plots in figure 5 and added the results in Table 1 for better understanding of the data.

  1. The size of Figure 5 should also be changed so that the presented data is clear, scaling used in Figure 6 would be more adequate. Figure 5 became figure 6. We changed the size of this figure to present the data clearly.

Reviewer 2 Report

In the present manuscript the authors describe how HCT116 spheroids interact with 5-FU after sensitizing them with prunus spinoza extract.

I do not recommend publication of the manuscript for the following reasons:

- The extract and the extraction process is very poorly defined.

- The preparation of the spheroids is poorly defined. Looking at the images, the authors were not able to prepare spheroids. Rather they made multicellular tube like constructs.

- The imaging is not well done. The points the authors claim you are not able to see on the images.

- The whole methods section needs reworking. You could not repeat the measurements the authors are describing without special knowledge.

- The discussion is a repetition of the results.

- There are no bars on the microscopy images in Fig 2 ...

- I am sorry, but: Your paper is entitled that you look at spheroids. You do not have spheroids.

Author Response

In the present manuscript the authors describe how HCT116 spheroids interact with 5-FU after sensitizing them with prunus spinoza extract.

I do not recommend publication of the manuscript for the following reasons:

- The extract and the extraction process is very poorly defined. We specified extraction protocol in the materials and methods section. The characterization of the compound was published in our previous work (Meschini et al., 2017).

- The preparation of the spheroids is poorly defined. Looking at the images, the authors were not able to prepare spheroids. Rather they made multicellular tube like constructs. We use nonstick flasks (code 4616/EX 3815), defined by the Corning company as suitable for making 3D spheroids (I attach the company reference). In addition, the resulting tubular appearance perfectly mirrors the structure of the colon carcinoma from which the cells are derived. Not all spheroids need to be round; the shape depends on the tissue of origin

- The imaging is not well done. The points the authors claim you are not able to see on the images.

The images presented in the article are from light microscopy, and the difference in membrane morphology of spheroid cells is well appreciated. However, we include the scanning microscope images that certainly show the difference between controls and treated.

- The whole methods section needs reworking. You could not repeat the measurements the authors are describing without special knowledge.  All methods have been comprehensively presented and therefore do not need modification.

- The discussion is a repetition of the results. We modify it according to what we think is right.

There are no bars on the microscopy images in Fig 2. We have added the missing bars even though there is an enlargement of the image in the text.

Round 2

Reviewer 2 Report

Literature reports (e.g. https://www.nature.com/articles/s41598-018-19384-0#Sec13, https://www.mdpi.com/1422-0067/16/11/26020, https://journals.plos.org/plosone/article?id=10.1371/journal.pone.0188100#sec002) show compact spheroid formation on ultra-low adhesive plates using HCT116 cells. I still think it is not proper to call the structures the authors have prepared spheroids, but this is mostly semantics.

The authors claim that “spheroids” after growth to approximately 100 µm were used. Figure 2 shows the morphology of the structures. Looking at the scale bar now provided, the structures are significantly larger than 100 µm. The authors need to discuss the apparent difference between their structures and spheroidal shapes that have been shown in literature (e.g. the papers mentioned above).

The authors should provide SEM images of the control to properly compare it with the treatments. The scale bars in Figure 4 are barely readable. 

Author Response

The authors claim that “spheroids” after growth to approximately 100 µm were used. Figure 2 shows the morphology of the structures. Looking at the scale bar now provided, the structures are significantly larger than 100 µm. The authors need to discuss the apparent difference between their structures and spheroidal shapes that have been shown in literature (e.g. the papers mentioned above).

The reviewer is right our spheroids are of varying size and shape. In the various works in the literature, non-homogeneous shapes are visible, this depends on the type of plates and the type of system used to obtain the three-dimensional structures.The tubular shape reflects very well the histotype from where they originate. The thing that comforts us and makes us confident in the result obtained is that certainly in a larger structure the treatment efficacy obtained is of greater interest than in smaller structures.

However, we enclose two works in which on the same HCT116 lines we observe images similar to ours both in fluorescence microscopy comparable to our light microscopy  (Fig:4 _ M.Zoetemelk et al.· 2019.Short-term 3D culture systems of various complexity for treatment optimization of colorectal carcinoma. Sci Rep. 2019 May 8;9(1):7103.)  and in SEM ( Fig. 2 in  Reducenti M., et al., , Biology (Basel). 2021 Sep 17;10(9):929. doi: 10.3390/biology10090929._The Ultrastructural Analysis of Human Colorectal Cancer Stem Cell-Derived Spheroids and Their Mouse Xenograft Shows That the Same Cells Types Have Different Ratios).

We modified the description of spheroid dimensions in materials and methods.

The authors should provide SEM images of the control to properly compare it with the treatments. The scale bars in Figure 4 are barely readable.

As requested by the reviewer, we added the SEM control in Figure 3 and modified manuscript text. In Fig. 4, we have improved the visibility of the bar.

Round 3

Reviewer 2 Report

The authors have addressed my concerns.